# RL2Grid: Benchmarking Reinforcement Learning in Power Grid Operations

## Abstract

Reinforcement learning (RL) has the potential to transform power grid operations by providing adaptive, scalable controllers essential for decarbonization and grid resilience. However, despite their promise, today's RL methods struggle to deal with complex dynamics, aleatoric uncertainty, long-horizon goals, and hard physical constraints, hindering their application in power grids and other real-world settings. In this work, we present RL2Grid, a benchmark representing realistic power grid operations that aims to foster the maturity of RL methods. This work builds upon Grid2Op, a power grid simulation framework developed by RTE France, to provide standardized tasks, state and action spaces, and rewards within a common interface, and thereby provide a common basis for monitoring and promoting progress. We evaluate and compare widely adopted RL algorithms across the increasingly complex grid settings represented within RL2Grid, establishing reference performance metrics and offering insights into the effectiveness of different approaches (including pure RL approaches and hybrid approaches incorporating heuristics). Our findings indicate that power grids present substantial challenges for modern RL, underscoring the need for novel methods capable of dealing with complex real-world physical systems.

## 1 Introduction

Power grids play a key role in efforts to combat climate change, necessitating a rapid transition to low-carbon energy and improved robustness against climate-induced extremes. This requires power grids to operate under increasing speed, scale, and uncertainty, due in large part to evolving supply and demand profiles resulting from the integration of variable renewable energy sources and distributed devices (Li et al., 2023b). This integration creates significant challenges for human operators of power grids (Marot et al., 2022b), further exacerbated by the limitations of traditional power system solvers in addressing realistic systems (Chauhan et al., 2023). Deep reinforcement learning (RL) offers a promising approach to reshaping power grid operations, having demonstrated impressive performance in simulated environments such as Atari and Starcraft (Mnih et al., 2013; Papoudakis et al., 2021). However, there remain many open challenges that impede the practical application of RL in real-world environments–such as dealing with complex dynamics and aleatoric uncertainty, learning long-horizon goals, and satisfying hard physical constraints. We argue that *power grids encompass many of these challenges, which are also open research questions in RL.* For these reasons, investigating realistic power grid tasks from an RL perspective could yield substantial benefits for both society and the RL research community. However, progress in relevant RL methodologies is hindered by a lack of standardized benchmarks that can help promote and monitor progress, identify bottlenecks, and develop insights to address real-world challenges.

To address this gap, we present RL2Grid, an RL benchmark representing realistic power grid operations. RL2Grid captures a diverse, standardized set of increasingly complex power grid tasks that entail dealing with the combinatorially large number of possible actions available in typical grid operations. These tasks are presented within a standard Gymnasium-based interface, alongside common rewards, state spaces, and action spaces, in order to provide a common basis for comparison. To ensure the realism of the tasks represented, we build this benchmark upon Grid2Op (Donnot, 2020), a well-regarded simulation framework for sequential decision-making in realistic power grids. We additionally provide a comprehensive empirical comparison of widely adopted RL algorithms on the tasks captured within RL2Grid. In particular, we evaluate several model-free RL algorithms,

Figure 1: Top: Example of a discrete topological action to address an overloaded line. Bottom: Example of a continuous re-dispatching action to address an overloaded line.

as they are frequently used in the literature either as baselines or building blocks for more complex approaches. RL2Grid extends the well-known CleanRL codebase (Huang et al., 2022) to include flexible configurations for algorithm implementation details. Additionally, we integrate a heuristic module facilitating the seamless incorporation of basic grid operations (e.g., line reconnection and idle actions) into the training loop of existing algorithms, which we confirm yields a drastic improvement in performance and sample efficiency across all RL algorithms.

Through RL2Grid, we aim to provide a launching point to foster the maturity of RL methods within real-world environments such as power grids, notably by providing realistic tasks that encompass important open questions, and by providing a standardized basis for comparative evaluation and analysis of paths forward. We further assess the effectiveness of popular learning approaches on the RL2Grid tasks. Finally, we discuss important open problems in power grids and their relationship to open problems in RL, as well as highlighting directions for further improving the realism of the power grid simulators, which is a necessary next step to enable last-mile development and deployment of the more general methodological advances we hope RL2Grid will promote.

## 2 PRELIMINARIES

### 2.1 REINFORCEMENT LEARNING FORMALIZATION

We consider power grid problems that can be defined as a Markov decision process (MDP), modeled as a tuple $(\mathcal{S}, \mathcal{A}, \mathcal{P}, \rho, R, \gamma)$; $\mathcal{S}$ and $\mathcal{A}$ are the finite sets of states and actions, respectively, $\mathcal{P} : \mathcal{S} \times \mathcal{A} \times \mathcal{S} \to [0, 1]$ is the state transition probability distribution, $\rho : \mathcal{S} \to [0, 1]$ is the initial uniform state distribution, $R : \mathcal{S} \times \mathcal{A} \to \mathbb{R}$ is a reward function, and $\gamma \in [0, 1)$ is the discount factor. In policy optimization algorithms, agents learn a parameterized stochastic policy $\pi : \mathcal{S} \times \mathcal{A} \to [0, 1]$, modeling the probability of taking an action $a_t \in \mathcal{A}$ in a state $s_t \in \mathcal{S}$ at a certain step $t$. We can also design value-based algorithms by defining state and action value functions $V_\pi$ and $Q_\pi$, which model the expected discounted return when starting from a state $s$ (and action $a$ for $Q_\pi$) and following the policy $\pi$ thereafter as:

$$V_\pi(s) = \mathbb{E}_\pi \left[ \sum_{t=0}^\infty \gamma^t R(s_t, a_t) | s_0 = s \right], \ Q_\pi(s, a) = \mathbb{E}_\pi \left[ \sum_{t=0}^\infty \gamma^t R(s_t, a_t) | s_0 = s, a_0 = a \right].$$

Given the current state and action, we can also measure how much better or worse the agent performs compared to its expected performance using the advantage function $A_\pi(s, a) = Q_\pi(s, a) - V_\pi(s)$. In these contexts, agents typically use a greedy policy over the action value or the advantage function (i.e., they take the action corresponding to argmax over the values). The goal is to find a policy that maximizes the expected discounted return.

### 2.2 SETTING: TOPOLOGY OPTIMIZATION AND RE-DISPATCH

RL2Grid considers the general setting of operating a power grid via topology optimization, as well as re-dispatch and curtailment actions, in order to keep the grid operational over a long horizon. To

clarify the setting our benchmark addresses, Figure 1 illustrates a simplified power grid scenario. This grid consists of four substations interconnected by transmission lines (edges), with two power generators and two loads connected to buses within each substation. Generators produce power to meet the demands (loads); the power flows through transmission lines, which also leads to power losses due to resistive heat on the lines; and substations (which may contain multiple buses) can act as "switches" to direct power flows to an extent. All of these electrical components have physical limitations; for instance, generators have ramping limits that prevent arbitrary instantaneous changes in power output, and transmission lines have maximum carrying capacities, with prolonged overloads potentially causing permanent damage and disconnections. RL has the potential to address such disruptions in real time, for instance by considering the two following categories of actions:

- Topology optimization (Figure 1 top) involves identifying substations where a bus-split action—the type of topological action we consider—can mitigate the overload by adjusting the grid topology (i.e., how elements are currently interconnected in the grid). This approach is cost-effective for grid operators as it typically involves simple switch activation.[1] However, determining the "optimal" topology from the combinatorial number of possible configurations is typically infeasible using existing optimization-based solvers.

- Re-dispatch or curtailment (Figure 1 bottom) deals with adjusting the power flow by re-dispatching or curtailing the power output of fossil and renewable power generators (respectively). However, this method is often economically demanding as it disrupts the normal operations of third parties controlling the generators and can lead to additional power costs.

## 2.3 GRID2OP

Grid2Op is an open-source simulator designed by RTE France (France's transmission system operator) to model sequential decision-making on a power grid (Donnot, 2020). It allows for the testing of various control algorithms, including RL policies, in relatively realistic scenarios. In particular, Grid2Op models important complexities in the power grid, including realistic non-linear dynamics, uncertainty deriving from time-varying renewable energy sources, and the massive amount (combinatorially large number) of grid configurations and actions that exist even in moderately-sized grids. It then presents various scenarios that simulate typical grid operations, requiring that grids are kept operational for long horizons in a way that is robust to contingency events (i.e., unexpected failures), as well as adhering to physical and operational constraints. Contingencies include transmission line disconnects, deterministic maintenance, and stochastic ("adversarial") events such as overloads (potentially caused by extreme weather conditions). With respect to operational constraints, Grid2Op imposes operational constraints such as: (i) cooldown periods to prevent immediate reconnection of disconnected lines, and limits on the frequency of actions on the same line to avoid asset degradation; (ii) limited thermal capacity of transmission lines; (iii) ramp rates on generators that restrict how much power generation can change between time periods; and (iv) adherence to AC power flow constraints. Moreover, using external packages like chronix2grid (Marot et al., 2020a), Grid2Op's scenarios include time series data that model load demands and generator outputs necessary to satisfy cumulative demand under ideal conditions without line capacity limits.

To date, Grid2Op has been primarily used as the computation engine for the "Learning to Run a Power Network" (L2RPN) competitions (Marot et al., 2020b; 2021; 2022a). While a number of RL-based methods have been proposed over the years for L2RPN, they fail to provide a common ground to foster advancements in RL methodologies. For example, each method employs different grid features as input for the agent and different action spaces of (very) limited size, often without providing sufficient evidence on how and why these action spaces were considered. For these reasons, to date, there is no standardized solution that allows RL researchers to easily get started in this field and compare over an established benchmark. To address this gap, our work builds on Grid2Op to provide a benchmark with standardized tasks, state and action spaces, and rewards, as well as comprehensive evaluation of strong baseline methods; these are critical to provide a common basis for assessing advances in RL methods (Papoudakis et al., 2021) as well as to improve accessibility to RL practitioners who may have limited prior knowledge of power systems. (See also Appendix A for further discussion on the relationship between L2RPN and our RL2Grid benchmark.)

---

[1]There is some uncertainty (and debate) regarding how frequently each component can be switched safely in practice, without degrading the underlying equipment.

Table 1: List of base environments currently supported by RL2Grid. For more details, see the original Grid2Op documentation (Donnot, 2020).

| ID | Maintenance | Opponent | Battery | # Subs. | # Lines | # Gens. | # Loads |
|---|---|---|---|---|---|---|---|
| **bus14** | ✓ | ✗ | ✗ | 14 | 20 | 6 | 11 |
| **bus36-M** | ✓ | ✗ | ✗ | 36 | 59 | 22 | 37 |
| **bus36-MO-v0** | ✓ | ✓ | ✗ | 36 | 59 | 22 | 37 |
| **bus36-MO-v1** | ✓ | ✓ | ✗ | 36 | 59 | 22 | 37 |
| **bus118-M** | ✓ | ✗ | ✗ | 118 | 186 | 62 | 99 |
| **bus118-MOB-v0** | ✓ | ✓ | ✓ | 118 | 186 | 62 | 91 |
| **bus118-MOB-v1** | ✓ | ✓ | ✓ | 118 | 186 | 62 | 99 |

## 3 RL2GRID BENCHMARK

In this section, we present RL2Grid, a benchmark for RL in power grid operations. We discuss the main features of the power grid environments presented as part of RL2Grid (which we wrap within a standardized Gymnasium interface), as well as our approach to standardizing the action and state spaces and the reward function for these environments in order to create a standardized set of tasks. We further discuss the set of widely-adopted RL algorithms that we assess as baselines on these tasks, as well as presenting a heuristic module that enables basic grid operations to be incorporated within the training loop of existing RL algorithms.

### 3.1 RL2GRID TASKS

The tasks presented within RL2Grid are designed on top of 7 main "base environments" from Grid2Op. Each of these "base environments" has a double bus system, meaning that every electrical component (i.e., generator and load) has two possible connections within a substation. Table 1 summarizes the base environments, including their features and the number of electric components. These environments include various types of contingencies such as

(i) Maintenance (M): Scheduled events that the agent is aware of (included in the state). During maintenance, a line is disconnected for a specified period and cannot be reconnected until maintenance is complete. (ii) Opponent (O): Unforeseen events (e.g., weather conditions) that cause a random line to disconnect (Omnes et al., 2021). The agent does not know about these events in advance and must react in real-time, using topological or re-dispatching/curtailment actions. Once a line is disconnected, it enters a *"cooldown"* state, during which it cannot be reconnected for several steps. Environments may also include storage units (batteries (B)) that can act as both generators and loads. Batteries can store a given amount of energy, which can be discharged as needed.

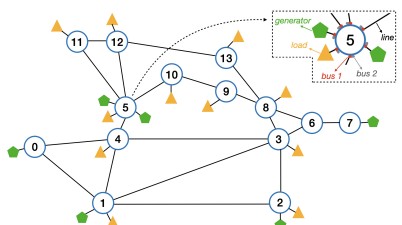

Figure 2: IEEE 14-bus sample grid.

**Action spaces.** For each base environment, we consider two types of tasks based on their action spaces, resulting in a total of 39 tasks.

(i) Topology (T): Agents can take discrete actions related to topological changes. These include disconnecting or reconnecting a line, or changing the bus to which an electrical component (load, generator, battery, or transmission line) is connected in a substation. These actions are virtually free since they only involve remotely activating a switch. However, they represent a significant challenge for grid operations, as the number of discrete actions scales exponentially with the number of elements connected to the substation. Human operators currently modify the grid topology manually based on historical behaviors; there is no tractable approach to obtain optimal topology optimization solutions (at scale) as of yet. As analyzed by (Chauhan et al., 2023), the number of topological re-configurations $N$ for a double bus substation comprising of $N_{\text{lines}}$ lines, $N_g$ generators, $N_l$ loads, is:

$$N = 2^{N_{\text{lines}}+N_g+N_l-1} - 2^{N_g+N_l} - 1.$$

For instance, substation #5 of the *bus14* grid (Figure 2) has 2 generators, 1 load, and 4 lines (7 elements), resulting in 55 possible actions. In more complex tasks like *bus36* and *bus118*, a single substation can have over 65,000 possible topologies.

Given the large discrete action space, we create different versions of the topology environments ("difficulty levels"), in which different numbers of topology actions are available to the agent.[2] We selected the action spaces for these difficulty levels through extensive simulations (48 hours on the computer cluster detailed in Section 4) in which we ranked the full set of discrete actions based on their *survival rate* for the grid. The survival rate represents how long the grid operates in normal conditions over an episode—the normalized number of steps for which an action does not cause a grid collapse (i.e., because the total demand is not satisfied or parts of the grid become disconnected). Specifically, we uniformly sampled actions from the topological space, and counted the number of times these actions did not cause a grid collapse over the simulation. After ranking the actions by survival rate, we take the first $N_{\text{actions}}$ from the ordered action space, where $N_{\text{actions}}$ differs at each difficulty level. Each increasing level of difficulty thus features a higher-dimensional discrete action space. Appendix C summarizes the difficulty levels with the corresponding total number of actions. Considering these levels, RL2Grid has a total of 32 topology-based environments. To motivate our action ranking method, we also visually analyze the resultant action spaces in Appendix C.1.

(ii) Redispatching and curtailment (R): Agents can take continuous actions related to costly dispatching changes. Costs arise from altering the planned generation schedule of power plants, increased fuel costs, and financial compensation for renewable energy producers, to name a few examples. Redispatching actions apply to fossil fuel-based generators, while curtailment actions apply to renewable energy-based generators. Batteries, if present, are also considered generators and add continuous actions for charging/discharging operations. This action space is relatively tractable for RL algorithms since it involves one continuous action per generator (i.e., $N = N_g$). Thus, we present a total of 7 continuous action-based training environments (one per base environment, in which all possible redispatching and curtailment actions are available to the agent).

**State space.** Agents have access to the state of the power grid at each time step. The state includes common grid features such as production at each generator, load demands, status, capacity, and cooldown of transmission lines, as well as the current step. Additional features are provided based on the environment's characteristics (e.g., maintenance, opponent events, batteries) and the action space. For example, in the topological case, the state includes the topological vector, the connection status of lines, overflow status, and substation cooldowns. In the continuous case, the state includes target and actual dispatches, curtailment, and generator ramping limits. An exhaustive list and description of the features that comprise the state is discussed in Appendix D.

**Reward.** The reward function is designed to encourage the agent to keep the grid operational for as long as possible while minimizing: (i) the capacity of the transmission lines (i.e., how much they are used), (ii) changes to the topology (only for the topology action space); (iii) costs related to re-dispatching operations. Appendix C.2 provides an exhaustive description of this reward function.

## 3.2 RL2GRID BASELINES

We assess the performance of a number of RL methods on the above tasks. In particular, we select a set of methods that are commonly used in the RL literature and serve as building blocks for more complex algorithms. These methods are discussed in more further depth in Appendix B.

**DQN** (Mnih et al., 2013) approximates the $Q$-function, using an $\epsilon$-greedy policy at training time. Due to its value-based nature, a DQN agent can only consider discrete (topological) actions.

**PPO** (Schulman et al., 2017) directly approximates a policy by learning its parameters using a computationally tractable clipped objective. By learning different probability distributions, a PPO agent can deal with continuous (re-dispatching) and discrete (topological) actions.

**SAC** (Haarnoja et al., 2018) uses different networks to learn a policy and two value functions that mitigate positive bias in value estimates. An SAC agent can deal with the same action types as PPO.

---

[2]We did not consider splitting the environment into difficulty levels for expensive re-dispatching actions, due to the limited size of this continuous action space.

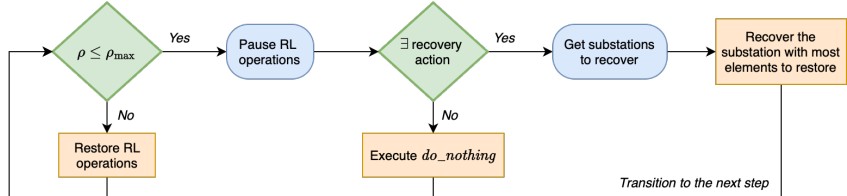

Figure 3: Recovery-based heuristic method designed for RL2Grid. When a line surpasses the capacity threshold, the RL agent picks actions to address the risky situations. In the other case, the heuristic computes the recovery actions to revert the current grid configuration to its original one. We operate on a maximum of one substation per each step, which resembles realistic grid operations.

**TD3** (Fujimoto et al., 2018) is similar to SAC, but uses multiple networks to learn a deterministic policy and can only deal with continuous actions.

### 3.3 HEURISTIC-GUIDED RL

Given the complexity of the topological actions, we introduce two baseline heuristics to assist RL agents' operations. These heuristic-guided approaches have been employed in various forms in previous work (Donnot, 2020; Marot et al., 2020b), but there is no widely adopted solution that: (i) can be easily used by RL practitioners, and (ii) has been benchmarked across different tasks to establish a standard for comparison and evaluation.

We refer to the first heuristic as "idle," as it does not perform any operation on the grid when line capacities are below the safety threshold of 95%. The second heuristic is a "recovery policy," (Figure G.3) as it restores the grid's original topology or performs idle actions when line capacities are below the safety threshold of 95%. In more detail, when line capacities exceed the safety threshold, the RL agent selects and executes an action based on the current state to bring the grid back to normal operation. When the grid operates normally (i.e., all line flows are under the safety threshold), the recovery policy takes over. If the grid is in its original topology, the heuristic performs an idle action to proceed with the simulation without changes. Otherwise, the heuristic calculates the actions needed to revert each substation to its original configuration. Considering realistic operational limits (i.e., change at most one substation per step), the heuristic first recovers the substation that is most different from its original configuration.

Importantly, the recovery actions do not disrupt RL agent training; instead, they emulate typical human operator behavior. Under normal conditions, system operators prefer to maintain the starting topology. Therefore, it is challenging for an RL agent to learn to restore the original topology in high-dimensional action spaces. For this reason, we anticipate that RL baselines augmented with the recovery policy will drastically improve performance and sample efficiency.

## 4 EXPERIMENTS

This section presents a comprehensive evaluation of the performance of DQN, PPO, SAC, and TD3 in 18 representative environments of our benchmark. In particular, we tested DQN, PPO, and SAC on the discrete topological action space for the *bus14, bus36-MO-v0, bus118-M, bus118-MOB-v0* over all levels of difficulty (i.e., with increasingly bigger action spaces). Moreover, we tested PPO, SAC, and TD3 in the continuous re-dispatching action space of these environments.

Our experiments address the following key questions: (i) Can common model-free RL methods deal with high-dimensional power network operations? (ii) What is the impact of integrating existing task-level knowledge as an heuristic-guided policy within these real-world tasks?

**Implementation Details.** Data collection is performed on Xeon E5-2650 CPU nodes with 64GB of RAM, using CleanRL-based implementations for the baselines (Huang et al., 2022). Complete hyperparameters are in Appendix F. Due to the considerable number of algorithms and environments considered, we report the average return smoothed over the last 500 episodes of 5 runs per method. Shaded regions represent the standard error. Additionally, we set a strict time limit on the nodes used for data collection, set to 36 hours. As such, different algorithms have different computational

Table 2: Average survival rate of the grid obtained by baseline RL algorithms (*Std.*) and their heuristic versions (*H.(N)* for the idle heuristic, *H.(R)* for the heuristic that restores the grid to its original topology) in representative environments (difficulty 0) with topological actions for DQN, PPO, SAC.

| | | DQN | | | PPO | | | SAC | | |
|---|---|---|---|---|---|---|---|---|---|---|
| Env. | Diff. | *Std.* | *H.(N)* | *H.(R)* | *Std.* | *H.(N)* | *H.(R)* | *Std.* | *H.(N)* | *H.(R)* |
| **bus14** | *0* | 0.07 | 0.39 | 0.45 | 0.46 | 0.95 | 0.94 | 0.04 | 0.19 | 0.15 |
| **bus36-MO-v0** | *0* | 0.09 | 0.14 | 0.19 | 0.12 | 0.17 | 0.29 | 0.08 | 0.10 | 0.13 |
| **bus118-M** | *0* | 0.06 | 0.17 | 0.18 | 0.07 | 0.13 | 0.18 | 0.15 | 0.18 | 0.19 |
| **bus118-MOB-v0** | *0* | 0.08 | 0.19 | 0.27 | 0.10 | 0.18 | 0.28 | 0.07 | 0.15 | 0.19 |

Table 3: Average survival rate of the grid obtained by baseline RL algorithms in representative environments with continuous re-dispatching actions (*Cont.*) for PPO, SAC, TD3.

| | | PPO | SAC | TD3 |
|---|---|---|---|---|
| Env. | Diff. | *Cont.* | *Cont.* | *Cont.* |
| **bus14** | *0* | 0.17 | 0.001 | 0.06 |
| **bus36-MO-v0** | *0* | 0.08 | 0.02 | 0.01 |
| **bus118-M** | *0* | 0.18 | 0.003 | 0.01 |
| **bus118-MOB-v0** | *0* | 0.25 | 0.08 | 0.07 |

requirements, and some of the baselines run for more time steps than others. For example, the heuristic-guided methods are much more computationally demanding than the baselines given that they often have to check and compute the reverting actions for the power network, and thus run for considerably fewer steps than the baselines.[3] Given the computational resources used, Appendix E addresses the associated environmental impact and our efforts to offset estimated $CO_2$ emissions.

## 5 RESULTS

Table 2 shows the preliminary results of our evaluation for topological action spaces. We indicate with *Std.* the original model-free baseline, and with *H.* the baseline augmented with the heuristics described in Section 3.3. For these topological cases, we report the results for the first level of difficulty (i.e., considering 50 discrete actions), and refer readers to the Appendix for the complete results and training curves. Table 3 shows the results for the re-dispatching action spaces.[4]

Overall, we notice that all model-free algorithms struggle to deal with the complexities of power network operations described in Section 2. Considering the lower number of training steps, we also notice that the heuristic-guided versions of the baselines typically achieve higher performance, despite being not nearly sufficient to operate the grid for long periods of time. These results further motivate the need for further advancements in RL algorithms that can contend with the complex dynamics and aleatoric uncertainty, long-horizon goals, and hard physical constraints represented within these tasks. By providing a common ground to the community, we hope to foster further research on these fronts.

## 6 RELATED WORK

There have been several attempts to develop benchmarks for sequential decision-making in power system operations, but they often focus on smaller-scale problems and/or simplified setups (Chen et al., 2022). Examples include python-microgrid for simulating microgrids (Henri et al., 2020), CityLearn for demand response and urban energy management (Vazquez-Canteli et al., 2020), and gym-ANM for active network management in small electricity distribution networks (Henry & Ernst, 2021). RL environments and algorithms for electric vehicle (EV) charging and electricity markets

---

[3]We refer to the appendices for exhaustive details about the training runs.

[4]We recall the continuous action space only has one level of difficulty since it only considers one action for each generator.

have also been introduced (Zhang et al., 2020). Recently, SustainGym spanned diverse tasks ranging from EV charging to carbon-aware data center job scheduling (Yeh et al., 2023). The ARPA-E GO Competition provides a realistic, large-scale benchmark for power grid operations (ARPA-E, 2023), but is more-so geared towards offline optimization approaches than online sequential decision-making. On the methodological side, recent contributions in the field include works on cascading failure mitigation, demand response optimization, and real-time grid control using RL (Matavalam et al., 2022; Lehna et al., 2023; van der Sar et al., 2024). Nonetheless, these works are more geared towards methodological advancements rather than proposing a benchmark. For this reason, we refer the reader to recent reviews for details on RL applications in power grid operations (Li et al., 2023b;a).

## 7 TACKLING THE CHALLENGES OF POWER GRIDS WITH RL

Applying RL in power grids presents numerous open problems, each offering significant opportunities for advancing both grid operations and RL methodologies (Marot et al., 2022b). While we address a subset of these challenges via our benchmark, there remains ample room for future work.

### 7.1 RL METHODOLOGIES OF IMPORTANCE FOR POWER GRIDS

Different RL techniques have the potential to be beneficial in addressing open problems in power grids. However, there are also potential risks – e.g., with respect to safety, reliability, and robustness – that are important to address. In the following, we summarize interesting avenues for future research to both realize the potential of RL in power grid operations and mitigate its risks.

**Safe RL.** Safety is paramount in power grid operations. Safe RL methods aim to ensure that learning and control policies adhere to strict safety constraints, preventing actions that could lead to blackouts or equipment damage (Donnot, 2020). Ensuring safety while optimizing performance is a critical area of research (García & Fernández, 2015). In particular, incorporating Constrained Markov Decision Process (C-MDP) representations, which explicitly handle constraints, can be particularly beneficial for ensuring that solutions adhere to physical and operational limits (Liu et al., 2021).

**Human-in-the-loop.** Effective grid management often requires human expertise and intervention. Incorporating human supervision, interaction, and feedback into RL systems allows for a synergistic approach where human operators and AI systems work together to optimize grid operations (Marot et al., 2022b). This collaboration can enhance decision-making and build trust in AI-driven solutions.

**Hierarchical control and multi-agent learning.** Power grids operate across multiple hierarchical levels, from individual substations to entire regions. Effective coordination within and across these levels is crucial for maintaining efficient and reliable grid operations. Hierarchical RL methods can be developed to manage these multi-level control tasks, in a way that addresses the scale and complexity inherent in grid operations (Pateria et al., 2021). Another promising direction is the use of multi-agent representations. Given the vast and distributed nature of power grids, scalability can be enhanced by dividing the grid into distinct areas or agents, each responsible for its own operations. Multi-agent RL (MARL) frameworks can enable these agents to learn and coordinate actions (Papoudakis et al., 2021), to improve overall grid performance while managing local contingencies more effectively.

**Robust RL.** The integration of renewable energy sources introduces significant variability and uncertainty into power grids, leading to non-stationary environments. RL algorithms need to adapt to these evolving dynamics to ensure stable and efficient grid operations despite fluctuating supply and demand profiles. Handling non-stationarity is thus a critical research direction (Moos et al., 2022).

**Model-based RL.** Model-based RL methods leverage models of the grid dynamics to improve learning efficiency and policy performance. These methods can provide more accurate predictions and better generalize across different scenarios, leading to faster and more robust solutions (Luo et al., 2022). Additionally, the AlphaZero algorithm, which combines tree search with deep learning, has shown remarkable success in games like chess and Go and could offer new strategies for handling complex, sequential decision-making tasks with high-dimensional spaces (Liu et al., 2023).

**Better representations.** Improving model representations for RL in power grids can also lead to more efficient learning and better policy performance. Leveraging Graph Neural Networks (GNNs) offers a potential avenue for advancement. Power grids can be naturally represented as graphs, with nodes representing buses and edges representing transmission lines. GNNs can effectively model these

structures, capturing the spatial and topological dependencies inherent in power grids. Integrating GNNs with RL algorithms can enhance the representation and learning of grid dynamics.

**Non-RL approaches.** While RL holds great promise, it is also essential to consider non-RL approaches such as optimization solvers, which are relevant particularly for problems with well-defined optimization objectives and constraints. In addition, exploring hybrid methods that combine RL with traditional optimization techniques can yield powerful tools for complex grid management tasks.

## 7.2 IMPROVING REALISM OF POWER GRID ENVIRONMENTS

While RL2Grid aims to promote initial advancements in RL methodologies of relevance to power grids, it is important to acknowledge that this is only a first step. Notably, building on these advancements to develop "last-mile" deployed solutions will require further improvements in the realism of power grid environments. We highlight several important directions in this regard.

**Scalability.** Realistic power systems akin to those managed by operators like RTE France, National Grid ESO, and 50Hertz may capture hundreds to thousands of buses. To ensure that RL solutions are applicable to real-world scenarios, improving the size and scale of grid environments is essential.

**Real data.** Grid2Op (and thus, RL2Grid) relies on realistic but synthetic data, which already provide significant challenges for RL. After scaling up RL to deal with the challenges provided by RL2Grid, future environments should address current privacy issues and publicly release real grid data to design to bridge the gap with real power grid operations.

**N-1 security.** In real operations, grid operators must ensure the system can withstand failure of any single component. Rather than modeling failures via random opponents, environments should handle this exhaustively and/or through adversarial agents tailored specifically to the method being tested.

**Topology vs re-dispatch.** Different grid operators handle the relationship between re-dispatch and topology optimization differently. Future benchmarks should reflect this heterogeneity in how different power grids are managed. Moreover, Grid2Op's current approach of disconnecting lines after unaddressed overloads does not fully capture real-world practices, where operators attempt to prevent overheating at all costs. Incorporating more realistic consequences for unaddressed overloads, such as system costs, can improve the fidelity of benchmarks. Additionally, grid operators cannot switch every element to every busbar, and there are limits on the number of connected components per substation. Reflecting these constraints can lead to more practical and applicable RL solutions. Storage assets also play an increasingly important role in grid operations. Future benchmarks should accurately model storage and clarify the extent of control grid operators have over these assets.

**Phase-shift transformers.** Phase-shift transformers, currently modeled as integer variables in the action space, should be represented more accurately to reflect their operational impact. Maintenance activities also vary significantly, with Type A involving physical presence at the site and Type B allowing remote interventions. Differentiating these types of maintenance activities in benchmarks can provide a more accurate representation of real-world constraints.

## 8 CONCLUSIONS

Power grids are essential in combating climate change, requiring a transition to low-carbon energy and enhanced resilience against climate-induced extremes. The integration of variable renewable energy sources introduces complexities and uncertainties in grid operations, posing significant challenges for human operators and traditional power system solvers. Our work aims to foster progress towards these challenges by introducing RL2Grid, a benchmark designed to bridge the gap between current grid management practices and methodological research in RL. RL2Grid provides a standardized interface for power grid environments, featuring common rewards, state spaces, and action spaces across a pre-designed set of diverse and complex grid tasks in order to provide a common ground for monitoring and promoting progress. We perform a comprehensive evaluation of the performance of DQN, PPO, SAC, and TD3 on RL2Grid tasks, including versions augmented with domain-informed heuristics aimed at improving performance and sample efficiency, and find that there is still significant room for improvement in the performance of these methods. By offering a standardized platform for RL research in the context of power grids, RL2Grid aims to accelerate algorithmic innovation towards improving power grid operations amidst the evolving challenges posed by climate change.

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

## A RELATIONSHIP OF RL2GRID TO L2RPN TASKS AND SOLUTIONS

In this section, we clarify the relationship of the tasks presented within RL2Grid, as well as the baseline methods evaluated, to the tasks and solutions presented within the Learning to Run a Power Network (L2RPN) competition series.

**Tasks.** RL2Grid employs all the main Grid2Op "base environments" (which are likewise employed in L2RPN). However, the solutions developed for L2RPN relied on different customized components. Every competition relied on different time series, making effective comparisons far from trivial. For these reasons, on top of the standardization proposed in our work, we have made some underlying changes to the base environments to better reflect the current and future challenges of RL research. Examples include (i) episodes with longer horizons (i.e., an RL2Grid episode models a month of grid operations, ∼8000 steps, compared to weekly episodes of most prior work); (ii) making the tasks as uniform as possible (i.e., by integrating curtailment operations in all Grid2Op tasks); (iii) enabling simulation steps inside the Gymnasium interface (a feature added in our code revision, which is not currently available in Grid2Op). These decisions were driven by our goal of ensuring that our benchmark is accessible, standardized, and provides a clear starting point for researchers who may not be familiar with the nuances of these competitions and power grids.

**Baselines.** Due to the different choices of input features and action spaces considered by different methods submitted to the L2RPN challenges, it was not possible to directly benchmark these specific methods on the RL2Grid tasks. However, the baselines chosen are representative of the methods submitted to past L2RPN competitions, in addition to representing commonly-used methods within the RL community as a whole. In particular, within the L2RPN submissions, a common approach was to incorporate heuristics. These heuristics varied significantly between methods and pushed us to design one that mimicked human operations in real grid operations. We developed this heuristic in collaboration with power system operators who have contributed to our work, incorporating fundamental insights from previous solutions while keeping the focus on standardization and benchmarking.

## B RL BASELINES

In this section, we briefly introduce the baseline RL algorithms employed in our evaluation, referring to the original papers for exhaustive details about these methods (Mnih et al., 2013; Schulman et al., 2017; Haarnoja et al., 2018; Fujimoto et al., 2018).

**DQN** (Mnih et al., 2013). A DQN agent uses a neural network to approximate the action value function $Q$ by taking as input the state of the environment and outputting $Q$-values for every possible action. During training, the agent uses an $\epsilon$-greedy policy to select random actions or follow the greedy policy on these $Q$-values, according to a linearly decaying probability $\epsilon$. The $Q$ network is thus updated to minimize the difference between predicted $Q$-values and a target derived from actual rewards and future $Q$-values. To deal with overestimation, we use Double-DQN (van Hasselt et al., 2016) and decouple action selection from action evaluation using a target $Q$ network. Due to its value-based nature, a DQN agent can only consider discrete (topological) actions.

**PPO** (Schulman et al., 2017). A PPO agent uses its neural network to directly approximate a policy. The agent learns the policy parameters by simplifying the TRPO (Schulman et al., 2015) algorithm, using a computationally tractable clipped objective. This clipping mechanism prevents large changes to the policy that could destabilize the training. At a high level, such a surrogate objective balances policy improvement and limits the divergence between policy updates. To drive the policy training, PPO also learns an advantage function to determine how much better (or worse) taking an action is compared to the expected value. By employing different probability distributions as a policy, a PPO agent can deal with both continuous (re-dispatching) and discrete (topological) actions.

**SAC** (Haarnoja et al., 2018). Similarly to PPO, a SAC agent learns different networks to maintain a policy and two value functions that mitigate positive bias in value estimates. Overall, the agent maximizes both the expected return and the entropy of the policy. The entropy term encourages exploration by promoting stochastic policies, which helps prevent premature convergence to suboptimal policies. In terms of actions, the SAC agent can deal with the same action types as PPO.

**TD3** (Fujimoto et al., 2018). A TD3 agent learns multiple networks similarly to SAC. However, unlike the stochastic policies learned by PPO and SAC, TD3 learns a deterministic policy and can only deal with continuous actions. To encourage exploration, the agent does not maximize the entropy of the policy but adds noise to the output of the policy network.

## C ENVIRONMENTS

As discussed in Section 3, here we introduce the different levels of difficulty for the topological-based environments, as well as the reward function employed in all the tasks. Each increasing level of task difficulty corresponds to a higher dimensional discrete action space. Table C.1 summarizes the difficulty levels and the corresponding total number of actions.

### C.1 ACTION SPACES ANALYSIS

In this section, we visually analyze the action spaces of one representative environment for each power grid size (i.e., bus14, bus36-MO-v0, bus118-M).

For each difficulty level, Figures C.1, C.2 and C.3 show the percentage of actions considered for each substation within the action space. The x-axis lists the substation IDs in descending order based on the number of available actions. The y-axis represents the ratio of actions used in the action space to the total number of available actions for each substation. Consequently, the highest difficulty level indicates that the action space includes all possible actions for all substations. Overall, this analysis suggests that the substation with the most electric components (i.e., the most possible topologies) is best suited to handle contingencies.

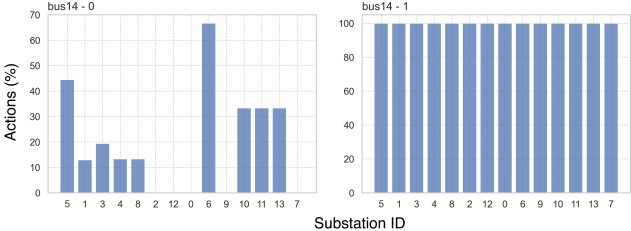

Figure C.1: Percentage of actions considered for each substation within the action space for bus14 (discrete) topological tasks (difficulty level is indicated with the number on the top left).

Table C.1: Action space sizes for the considered environments. Left: Difficulty for environments with a (discrete) topology-based action space. Right: (continuous) re-dispatching and curtailment tasks.

|  | *# Actions per difficulty level* | | | | | |
| --- | --- | --- | --- | --- | --- | --- |
|  | **Topology (T)** | | | | | **Redispatching and curtailment (R)** |
|  | *0* | *1* | *2* | *3* | *4* | *0* |
| **bus14** | 50 | 209 | - | - | - | 6 |
| **bus36-M** | 50 | 302 | 1829 | 11071 | 66978 | 22 |
| **bus36-MO-v0** | 50 | 302 | 1829 | 11071 | 66978 | 22 |
| **bus36-MO-v1** | 50 | 302 | 1829 | 11071 | 66978 | 22 |
| **bus118-M** | 50 | 308 | 1903 | 11744 | 72461 | 69 |
| **bus118-MOB-v0** | 50 | 309 | 1914 | 11849 | 73328 | 69 |
| **bus118-MOB-v1** | 50 | 309 | 1915 | 11852 | 73357 | 69 |

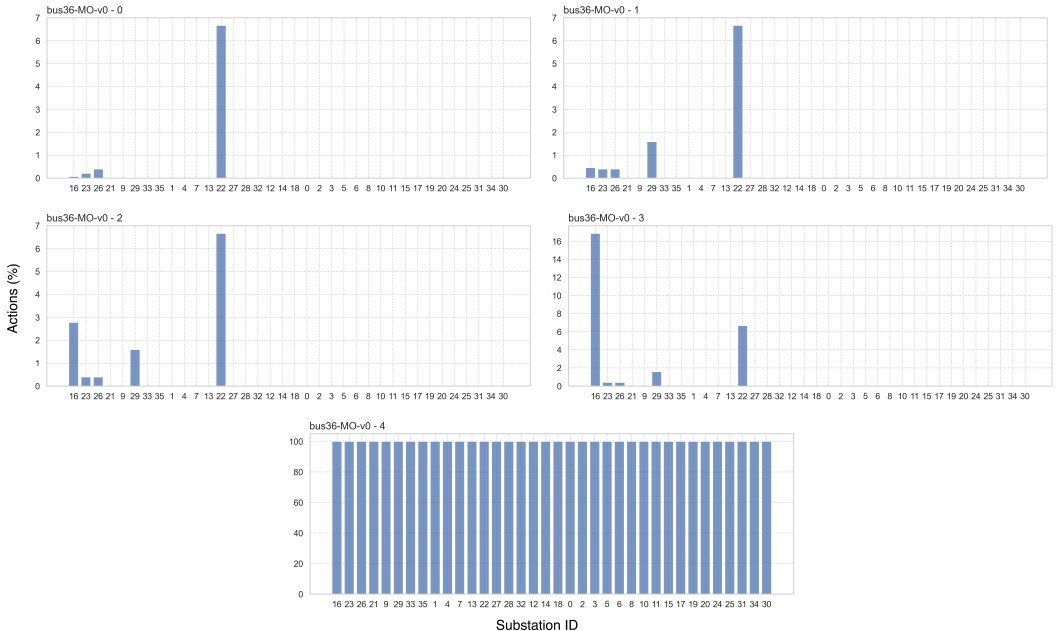

Figure C.2: Percentage of actions considered for each substation within the action space for bus36-MO-v0 (discrete) topological tasks (difficulty level is indicated with the number on the top left).

Figures C.4 and C.5 then presents the data collected during the action ranking mechanism described in Section 3.

As a sanity check, Figure C.4 shows an example of the uniform sampling strategy used to select which action to simulate at each simulation step. The x-axis shows the total number of actions for the bus14 (discrete) topological task; the y-axis indicates the number of times each action was sampled during the ranking process.

Figure C.5 shows the final ranking of the actions for the three representative environments. The x-axis shows the total number of actions for each task; the y-axis indicates the average survival rate of each action during the ranking process. Crucially, most of the actions are relevant (i.e., with a high survival rate) in the tasks, motivating the increasing levels of difficulty we proposed for the (discrete) topological environments.

## C.2 REWARD

To promote the survival of the grid, the agent gets a cumulative positive constant $R_{\text{survive}}$ for each step, normalized by the total length of a training episode (normalized $\in [0, 1]$). The capacity reward $R_{\text{capacity}}$ is based on how many lines are used (the lower the better and goes negative in case of overflow) and is set to a fixed penalty value for disconnected lines (normalized $\in [-1, 1]$). The costs component $R_{\text{cost}}$ assigns a cost to re-dispatching action and penalizes energy losses (normalized $\in [-1, 0]$). Finally, the topology component $R_{\text{topology}}$ incentivizes the agent to revert to the original topology by computing the distance of the current grid to the one at time 0 (normalized $\in [0, 1]$). The total reward $R$ an agent gets at each step is then a weighted sum:

$$R = \alpha R_{\text{survive}} + \beta R_{\text{capacity}} + \eta R_{\text{cost}} + \begin{cases} \omega R_{\text{topology}} & \text{if topology actions,} \\ 0 & \text{otherwise.} \end{cases}$$

## D STATE SPACE

Regardless of the task, at a certain time-step $t$ an agent gets the following set of features: $[t, \text{Gen}_P, \text{Gen}_\theta \text{Load}_P, \text{Load}_\theta, \rho, \text{Cooldown}_{\text{lines}}]$. Additionally, based on the nature of the task, the agent can observe additional features as follows:

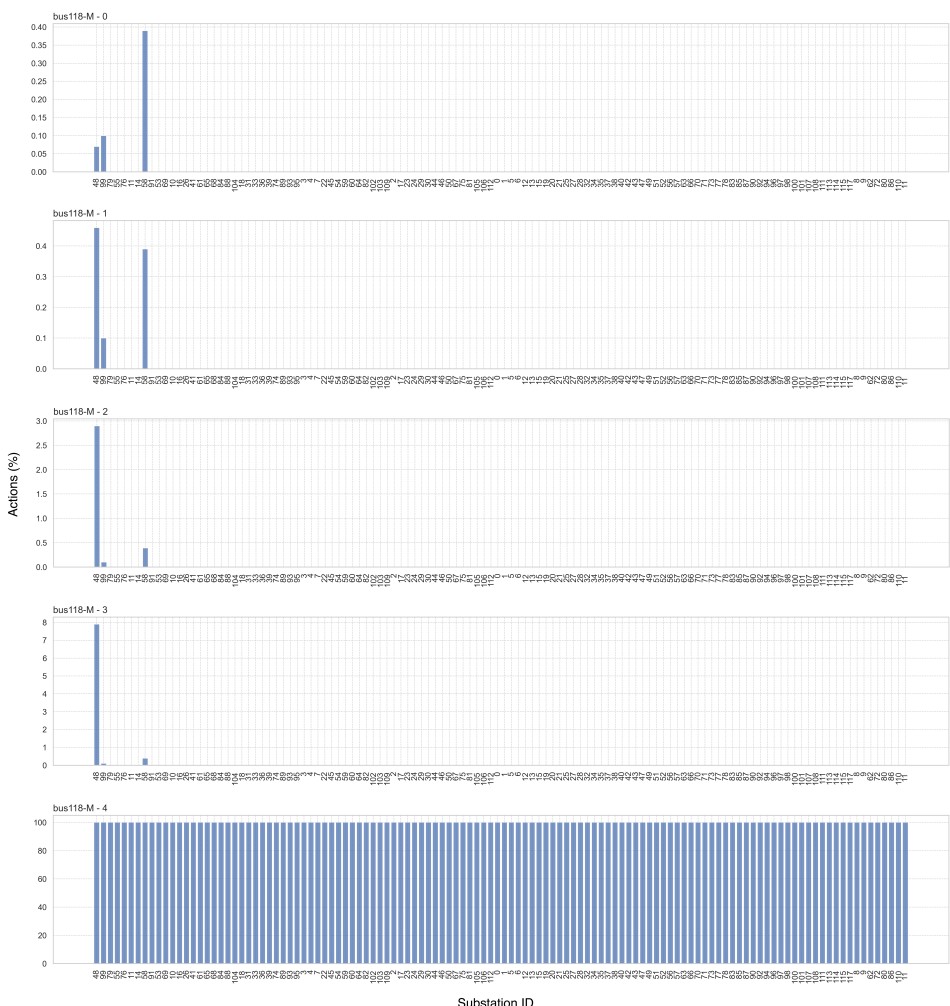

Figure C.3: Percentage of actions considered for each substation within the action space for bus118-M (discrete) topological tasks (difficulty level is indicated with the number on the top left).

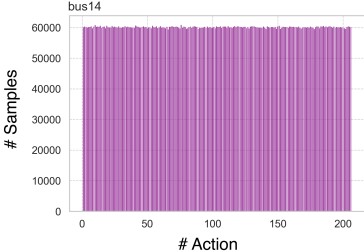

Figure C.4: Number of times each action is sampled over the ranking process time.

- Topological actions: when agents operate using (discrete) topological actions, it observes [$\text{Topo}_{\text{vect}}$, $\text{Line}_{\text{status}}$, $\text{Time}_{\text{overflow}}$, $\text{Time}_{\text{sub-cooldown}}$].

- Re-dispatching actions: when agents operate using (continuous) re-dispatching actions, it observes [$\text{Tg}_{\text{dispatch}}$, $\text{Curr}_{\text{dispatch}}$, $\text{Gen}_{\text{margin-up}}$, $\text{Gen}_{\text{margin-down}}$].

- Curtailment actions: when agents operate using (continuous) curtailment actions, it observes [$\text{Gen}_{\text{P}_{\text{curt}}}$, $\text{Curtail}$, $\text{Curtail}_{\text{limit}}$].

- Maintenance: when the task has maintenance contingencies (see Table 1), the agent gets [$\text{Time}_{\text{next-maint}}$, $\text{Duration}_{\text{next-maint}}$].

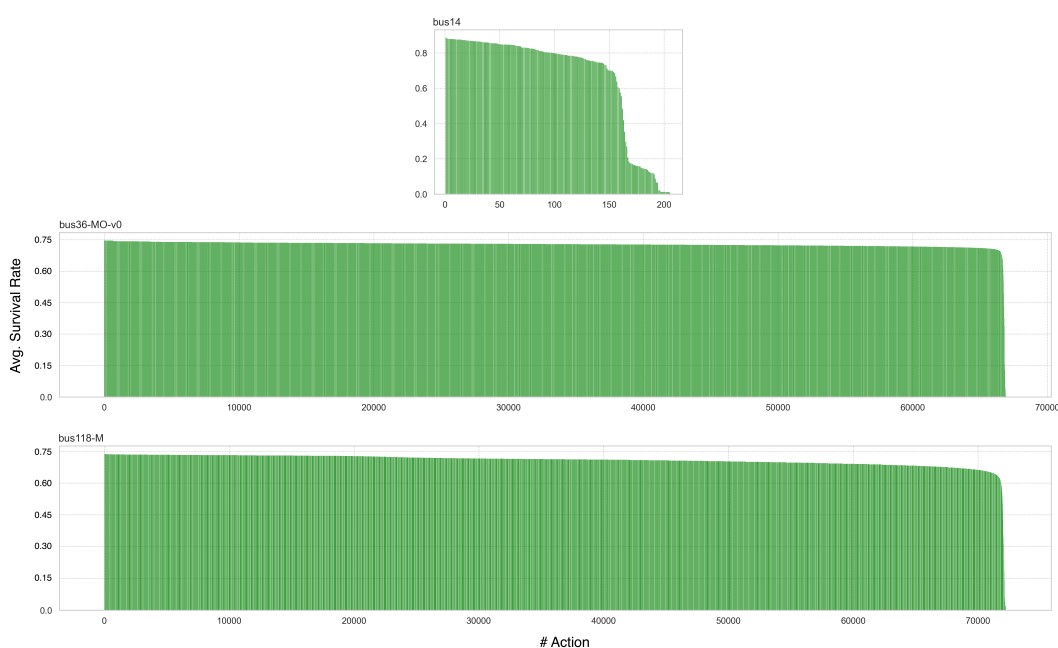

Figure C.5: Average survival rate of the action spaces after the ranking process time.

- Storage: when the task has batteries (see Table 1), the agent gets $[\text{Storage}_{\text{charge}}, \text{Storage}_{\text{power}_{\text{tg}}}, \text{Storage}_{\text{power}}, \text{Storage}_\theta]$.

Such a distinction is useful to reduce the size of the space the agent can observe when there are features that are not relevant to a specific task. For example, if an agent uses only discrete actions (topology) then everything related to target dispatch, actual dispatch, and storage is irrelevant as they will not change. Likewise, if an agent uses only continuous actions, it is not necessary to include features related to "topology" as they will not be modified. Additionally, all the features related to voltage (e.g., voltage for generators, loads, . . . ) and reactive values (e.g., reactive power for generator, loads, . . . ) can be neglected.

For the interested RL practitioner, we refer to the original Grid2Op documentation for exhaustive descriptions of these features (Donnot, 2020).

## E  ENVIRONMENTAL IMPACT

Despite each training run being "relatively" computationally inexpensive due to the use of CPUs, the experiments of our evaluation led to cumulative environmental impacts due to computations that run on computer clusters for an extended time. Our experiments were conducted using a private infrastructure with a carbon efficiency of $\approx 0.275 \frac{\text{kgCO}_2\text{eq}}{\text{kWh}}$, requiring a cumulative $\approx$720 hours of computation. Total emissions are estimated to be $\approx 20.79\text{kgCO}_2\text{eq}$ using the Machine Learning Impact calculator, and we purchased offsets for this amount through Treedom. We do not directly estimate or offset other categories of environmental impacts (such as water usage or embodied hardware impacts), though recognizing that these are additionally important to consider.

## F  HYPERPARAMETERS

Table F.1 lists the hyperparameters considered during our initial grid search and the final (best-performing) parameters used for our experiments.

Table F.1: Details of the grid search used to find the best-performing hyperparameters for each algorithm in the topological (T) and re-dispatching (R) cases.

| Algorithm | Parameter | Grid search | Chosen value (T - R) |
|---|---|---|---|
| **Shared** | *N° parallel envs* | 10 | 10 |
| | *Learning starts* | 20000 | 20000 |
| | *Time limit* | 48 hours | 48 hours |
| | *Max gradient norm* | 10, 20, 50 | 10 |
| | *Discount $\gamma$* | 0.9, 0.95, 0.99 | 0.9 |
| | *Batch size* | 64, 128, 256 | 128 |
| | *$\alpha$* | 0.1, 0.3, 0.6 | 0.1 |
| | *$\beta$* | 0.1, 0.3, 0.6 | 0.3 |
| | *$\eta$* | 0.1, 0.3, 0.6 | 0.3 - 0.6 |
| | *$\omega$* | 0.1, 0.3, 0.6 | 0.3 |
| **DQN** | *Train frequency* | 20, 100, 1000 | 20 |
| | *Target network update* | 500, 1000, 10000 | 1000 |
| | *Buffer size* | 100000, 250000, 500000, 1000000 | 500000 |
| | *Learning rate* | 0.003, 0.0003, 0.00003 | 0.0003 |
| | *$\epsilon$-decay fraction* | 0.3, 0.5 0.7 | 0.5 |
| **PPO** | *N° steps* | 10000, 20000, 50000 | 20000 |
| | *N° update epochs* | 20, 40, 80 | 40 |
| | *Actor learning rate* | 0.003, 0.0003, 0.00003 | 0.0003 - 0.00003 |
| | *Critic learning rate* | 0.003, 0.0003, 0.00003 | 0.0003 - 0.00003 |
| | *$\epsilon$-clip* | 0.1, 0.2, 0.3 | 0.2 |
| **SAC** | *Train frequency* | 20, 100, 1000 | 20 |
| | *Actor delayed update* | 2, 4 | 2 |
| | *Noise clip* | 0.5 | 0.5 |
| | *Buffer size* | 100000, 250000, 500000, 1000000 | 500000 |
| | *Actor learning rate* | 0.003, 0.0003, 0.00003 | 0.0003 - 0.00003 |
| | *Critic learning rate* | 0.003, 0.0003, 0.0003 | 0.0003 - 0.00003 |
| | *Entropy regularization* | 0.2 | 0.2 |
| | *Noise clip* | 0.5 | 0.5 |
| **TD3** | *Actor delayed update* | 2, 4 | 2 |
| | *Buffer size* | 100000, 250000, 500000, 1000000 | 500000 |
| | *Actor learning rate* | 0.003, 0.0003, 0.00003 | 0.0003 - 0.00003 |
| | *Critic learning rate* | 0.003, 0.0003, 0.00003 | 0.0003 - 0.00003 |
| | *$\tau$* | 0.005, 0.0005 | 0.005 |
| | *Policy noise* | 0.2 | 0.2 |
| | *Exploration noise* | 0.1 | 0.1 |

# G   OMITTED FIGURES IN SECTION 5

Figure G.1 shows the training curves for the (discrete) topological action spaces. Due to the strict time limit imposed on the computation nodes (see Section 4) and the different computational requirements of the algorithms, not all the baselines perform the same number of steps in the time limit.[5] Additionally, despite the grid search of Table F.1, some baselines achieved lower performance than expected (e.g., SAC and DQN in the *bus14* scenarios). We will keep working on the benchmark to find better parameters, run longer experiments, and keep the following Figures updated.

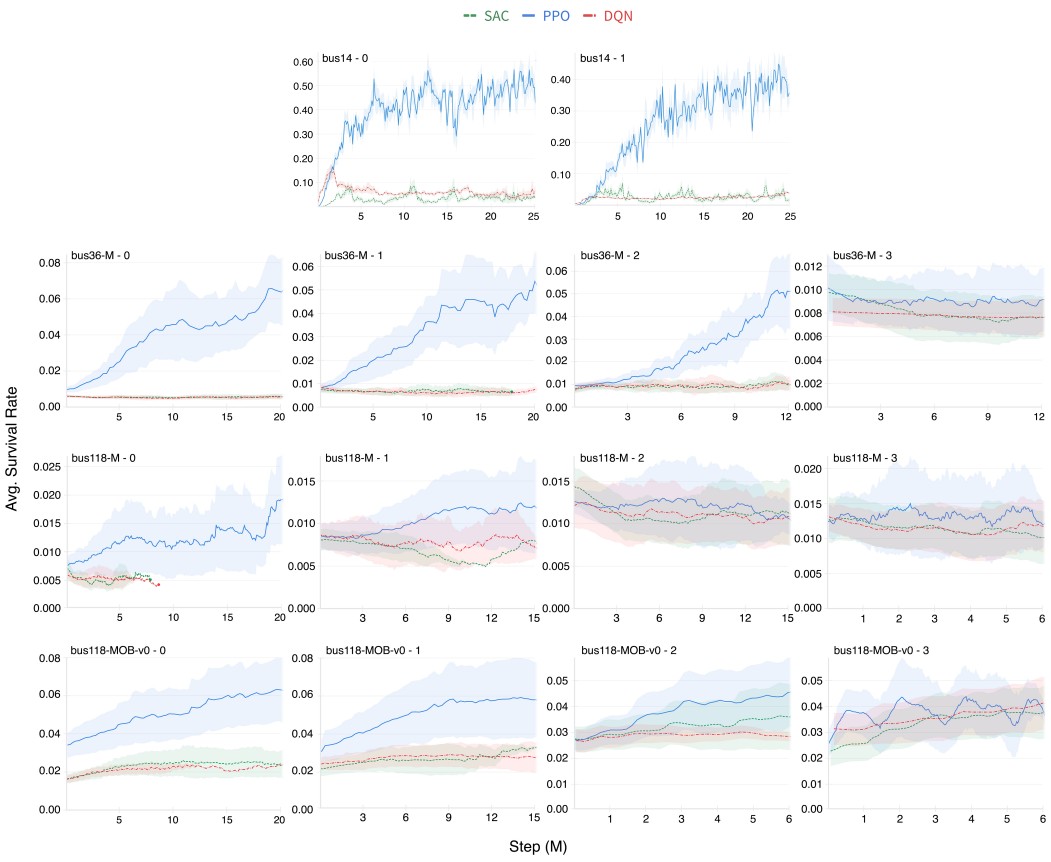

Figure G.1: Average survival rate for the discrete topological case in *bus14, bus36-M, bus118-M, bus118-MOB-v0* using the SAC, PPO, and DQN baselines. We indicate the difficulty level (ranging from 0 to 3) next to the environment identifier.

---

[5]The demands and limited performance of the topological baselines led us to exclude the results with the complete action space (i.e., difficulty set to 4).

Figure G.2 shows the training curves for the (continuous) re-dispatching action spaces.

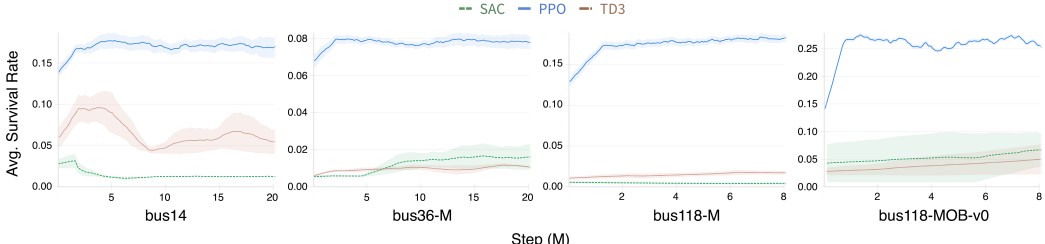

Figure G.2: Average survival rate for the continuous re-dispatching case in *bus14, bus36-M, bus118-M, bus118-MOB-v0* using the SAC, PPO, and TD3 baselines.

Figure G.3 shows the training curves for the recovery (R) and idle (I) heuristics applied to our topological bus14 scenario.

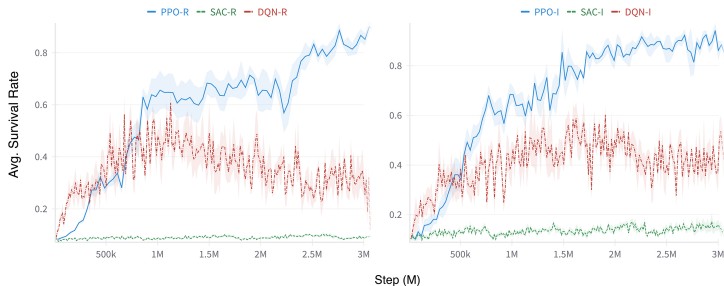

Figure G.3: Average survival rate for the discrete topological case in *bus14* using the PPO, SAC, and DQN baselines with the recovery (R) and idle (I) heuristics.

