# OpenReview forum: "RL2Grid: Benchmarking Reinforcement Learning in Power Grid Operations"
_ICLR.cc/2025/Conference — ICLR 2025 Conference Withdrawn Submission_

### Official Review · Reviewer_EBEa · 2024-10-31

**Soundness:** 1
**Presentation:** 2
**Contribution:** 2
**Rating:** 3
**Confidence:** 4

**Summary:**

This paper presents a benchmark for reinforcement learning (RL) in optimal power flow. Their work relies on Grid2OP, an existing framework generally used for RL and model predictive control (MPC) research. They analyzed the case from two points of view: finding balance through topology changes and re-dispatching/curtailment. Their approach aims to simplify the difficulty of the learning process by creating fractions of the action space (for the topology changes action) that are increasingly more challenging based on a metric they labeled survival rate, which expresses how long the grid operates in normal conditions over an episode without causing a grid collapse. Regarding re-dispatching/curtailment, they didn't add anything different from Grid2Op. They tested their framework with out-of-the-box implementations of state-of-the-art RL algorithms and reported the performance based on the survival rate.

**Strengths:**

The vision of proposing a curriculum-learning-like approach as a benchmark to RL in optimal power flow seems relevant for the research as it fosters the development of more meaningful contributions in the domain. From experience, I noticed that research in the domain tends to be disconnected from one another, lacking solid and uniform scenarios against which to compare. Including heuristics seems appropriate for OPF, considering that RL will never be able to completely replace the existing methods, so a fall-back policy is realistic. You considered the carbon impact of your research.

**Weaknesses:**

- In line 89, when you talk about MDPs, you formalize your RL case with finite states and actions that don't correspond to the action space of the re-dispatch/curtailment action.
- In line 90, you formalized S (the initial state distribution) as a uniform distribution, which seems unrealistic and different from the actual case you want to represent.
- In line 220, you reference a footnote, which I found imprecise. You talk about a limited size of a continuous action space. That is not the case, considering that you can consider infinite values in a continuous action state.
- In line 223, your primary metric should be clearly explained mathematically, especially if it's something you are introducing.
- Your main contribution, which is the creation of difficulty levels for topological action, needs to be explained clearly. It is not enough to say that you sampled uniformly because it needs to include the actual characteristics of the problem: you do sequential actions, advance in through states, and the Markov condition influences your action for the next state.
- In line 322, you mentioned your experiment setup in terms of runs, but for reproducibility, one expects to know which seeds you used to obtain your results, which you don't mention in the paper.
- In line 359, your conclusion seems to differ from what is expected from a benchmark for RL. Model-free methods are expected to struggle in scenarios like these, but you could've tuned them to be their best in your scenario.
- Instead of section 7, you could have spent more time explaining the details of your methodology.

**Questions:**

- What is the exact way you computed the survival rate?
- Could you please elaborate on the process of fractioning the action space?
- Did you sample from each node independently? If so, don't you think that there is a causal effect in the conditions of a line (edge) between two nodes (generator/load) if you decide to disconnect it randomly?
- What is your contribution to the re-dispatch/curtailment case?
- Why do you want to encourage the agents to return to the initial state through the reward function?

---

### Official Review · Reviewer_xuQW · 2024-11-02

**Soundness:** 2
**Presentation:** 3
**Contribution:** 2
**Rating:** 3
**Confidence:** 4

**Summary:**

This work is based on the power grid simulation framework Grid2Op developed by RTE France and proposes a benchmarking method called RL2Grid. It designs standardized state, action, and reward mechanisms, facilitating the testing and evaluation of various forms of RL-based grid optimization methods through a unified interface. This contributes to the development of more effective and reliable grid operation strategy models, helping to solve complex grid operation optimization problems in the real world, thereby bringing high cost-benefit ratios and providing quality grid services.

**Strengths:**

* The article introduces RL2Grid, a standardized evaluation platform that allows for the assessment and comparison of different reinforcement learning (RL) algorithms. It reveals the practical gap between current grid management practices and the application of RL methods, demonstrating the authenticity and reliability of the platform’s evaluation.

* The grid optimization scheduling problem and the evaluation of application methods discussed in the article involve a combination of research fields such as power systems, RL method optimization, and intelligent optimization scheduling, which hold significant economic and social research value.

**Weaknesses:**

* The article compares relatively simple reinforcement learning methods and only lists experimental results, failing to effectively analyze the reasons why these methods did not perform as expected or propose truly viable research solutions. Additionally, there is a lack of specific experimental results for rule-based methods, making it difficult to effectively assess the actual difficulty of the current environment.

* The article is based on the Grid2Op framework, which already includes methods and solutions from the L2RPN competition (e.g., “WINNING THE L2RPN CHALLENGE: POWER GRID MANAGEMENT VIA SEMI-MARKOV AFTERSTATE ACTOR-CRITIC”). However, these efficient design methods were not applied within the framework for a unified comparative analysis, which prevents the article from effectively showcasing the real progress in grid optimization research methods.

* Grid optimization is inherently a problem of target optimization. The article does not provide sufficient technical details or theoretical analysis to demonstrate the necessity of using reinforcement learning methods to solve such problems. It is unclear whether current methods, such as evolutionary computation, solvers, or even heuristic rules, have already achieved satisfactory results.

**Questions:**

* The poor performance of the RL baselines presented in the article might suggest that RL methods themselves are not suitable for handling such complex optimization problems. If RL does have the potential to solve these complex optimization problems, please provide specific examples or methodological designs to demonstrate this.

* RL2Grid is built on synthetic data from Grid2Op, but there is still a discrepancy between this and real grid operations. Is it possible to consider using real grid data for practical evaluation in the future?

* Could the article compare the proposed framework with existing grid evaluation frameworks and provide a table highlighting the comprehensive design points and advantages of the proposed benchmark?

---

### Official Review · Reviewer_yXNa · 2024-11-03

**Soundness:** 2
**Presentation:** 2
**Contribution:** 2
**Rating:** 3
**Confidence:** 4

**Summary:**

RL2Grid is a benchmark for RL algorithms, leveraging *Grid2Op* simulator to provide standardized tasks with a specific configuration of observation space, action space, and reward, to foster the application of new RL methods within a realistic power grid scenario. Authors evaluate and compare well-known RL algorithms across increasingly complex tasks to provide reference benchmarks.

**Strengths:**

- **Concept:** the benchmark on a complex realistic setting enables the study of new RL methods.
- **Framework:** the work is based on a well-structured simulator, accurately described by authors. Moreover, the integration of *Grid2Op* with *Gym* and *CleanRL* codebase foster the standardization of RL methods on power grid scenarios.
- **Literature:** authors provide several related works, proving the need for a benchmarking framework.

**Weaknesses:**

- **The paper is a bit chaotic and not well-organized.** For example, authors dedicate some sections in the main paper to explain basic concepts (section 2.1) and future works and improvements (section 7), which could have been left to appendix. Indeed, I would have rather given space to other important information, such as the MDP formulation of the actual problem, with a precise description of action and state spaces, and reward function, also reporting the ranges comprising such variables. Moreover, I would have put in the main paper also some plots, since from a benchmarking framework I expect to be provided with performance insights, corroborated with ideas on how the problem is tackled by baseline agents and the intuition behind their behavior.
- While authors present the proposed tasks giving a quantitative insight about the complexity of each task, **the MDP formulation is not appropriately tackled**. Indeed, authors describe the state space as a list of variables without explicitly explaining their meaning. Even if for some of them the reader can intuitively grasp what they represent, for others this is not so clear (Appendix D). Moreover, there is a lack of formalization about problem variables and their ranges. Finally, the reward function (appendix C.2) is not well-explained. For example, it is not explained how the components $R_{cost}$ and $R_{topology}$ are formally computed.
- **The paper does not bring significant novelty to the research.** The work is presented as a benchmarking tool for power grid, but in terms of contributions it seems to be just a mixing of different libraries (*Grid2Op*, *Gym*, and *CleanRL*). While the topic is interesting, the evaluation of state-of-the-art RL algorithms is not sufficient to build a benchmarking framework, lacking of formal KPIs to evaluate the goodness of each method and comparisons with non-RL baselines (even random or rule-base strategies), to testify the need of adopting RL approaches for this problem.

**Questions:**

- I did not understand why the discount factor $\gamma$ is considered in the grid search (Appendix F), since $\gamma$ is not a hyperparameter of the problem, but rather a part of the MDP that defines the problem, thus definitely not something to tune.
- While scenarios are carefully described in terms of complexity due to the increasing number of actions, I wonder if the task's difficulty would also depend on employed data: it would be nice to have an analysis of time-series used for experiments. Moreover, I did not understand the time step at which the simulator and agents work.
- Have you tried experimenting with longer episodes? Depending on the employed time step, 48 hours could be not sufficient to properly assess realistic scenarios. For example, power generation and consumption data significantly change in different seasons, and consumption data in particular also between work-days and weekends.
- Line 204: *"For each base environment, we consider two types of tasks based on their action spaces, resulting in a total of 39 tasks"*. From this sentence, I expect to be provided with an analysis of all the proposed tasks, at least in the appendix. Instead, it seems that the presented scenarios are fewer.
- Can you please provide some intuition behind the low performances of SAC and DQN with respect to PPO across the evaluated scenarios?
 - Regarding the third highlighted weakness, I would like to understand if you compared RL solutions to non-RL ones to testify the need for adopting such an approach to this problem and which are the metrics (KPIs) used in this evaluation.

---

### Official Review · Reviewer_3RQB · 2024-11-04

**Soundness:** 2
**Presentation:** 2
**Contribution:** 2
**Rating:** 5
**Confidence:** 4

**Summary:**

The paper presents RL2Grid, a benchmark framework for reinforcement learning (RL) methods in complex, real-world power grid operations. RL2Grid introduces standardized environments, tasks, and rewards for evaluating RL algorithms in power grid scenarios, such as topology optimization and power re-dispatching. It tests popular RL methods (e.g., DQN, PPO, SAC) alongside heuristic enhancements, revealing challenges in grid management and highlighting areas for improvement in RL’s application to power grids.

**Strengths:**

1. Introduces a real-world problem for RL, which is essential for advancing practical RL applications. More benchmarks like this could enhance RL’s potential for real-world deployment.

2.The paper effectively introduces a complex environment in a way that is easy for non-experts to understand.

**Weaknesses:**

1. The novelty of the paper seems somewhat limited. Since Grid2Op already existed, the main contributions appear to be the standardization of the action, state spaces, and reward function, along with some baseline testing. However, there’s a lack of explanation of what was actually standardized and why these specific definitions were chosen. It would be helpful to know how the environment functioned before standardization and what improvements were achieved through the new, standardized tasks, state and action spaces, and rewards.

2. The baselines included are limited. It seems that the L2RPN competitions previously addressed similar tasks, but this paper does not evaluate algorithms from those competitions.

3. There’s a lack of analysis on the results. It would be helpful for new researchers if the paper discussed why existing algorithms perform poorly and suggested insights on potential improvements.

**Questions:**

1. What exactly is the goal for agents in the RL2Grid environment? The performance tables compare agents using survival rate, so does this mean the agent’s primary objective is to survive as long as possible without termination? Are there other metrics to optimize ?

2. The reward design is confusing, specifically the minimization of transmission line capacity. Why is it beneficial to minimize the number of lines in operation? Is this a desirable objective?

3. What are the termination conditions for agents? It would be helpful if this was explained in Section 3.1.

4. What distribution governs the changes in generators and loads?

5. The environment includes two main actions: topology optimization and re-dispatching. Wouldn’t using both simultaneously be more effective? Why were there no experiments with agents using both actions?

6. Given the large action space, is it feasible to approach this problem with a single agent? It seems more practical to treat it as a multi-agent problem. Has any research applied a multi-agent approach to this problem?

---

### Note · Authors · 2024-11-20

**Comment:**

We sincerely thank the reviewers for their thoughtful feedback and valuable suggestions. After careful consideration, we have decided to withdraw our work to focus on enhancing the clarity of our contribution. However, we believe there are several points where misunderstandings may have arisen, and we would like to address them in the following brief discussion.

Our primary contribution is the development of RL2Grid, a comprehensive benchmark built on top of the existing Grid2Op environments. The distinction between a simulation environment and a benchmark is crucial: while Grid2Op provides a flexible and highly customizable environment, RL2Grid introduces a structured and reproducible suite of tasks, metrics, and baseline methods, offering a common basis for comparing RL algorithms under equivalent conditions. This standardization is key for facilitating meaningful insights into the strengths and weaknesses of various algorithms, which has been a challenge in previous works due to the high degree of customization allowed by Grid2Op.

Specifically, the key contributions of RL2Grid are related to:
- *Standardization*: Previous efforts, such as the L2RPN challenge series, have been valuable but often relied on highly customized setups, making it difficult to draw general conclusions. RL2Grid addresses this by providing a standardized benchmark that allows researchers to compare methods on a level playing field. While it is true that state and action spaces can be designed on top of Grid2Op, our approach drastically simplifies this process for the broader research community by offering pre-designed, well-tested configurations. This design choice combats the entry barrier for researchers new to the field of power systems and those who wish to focus on algorithmic development rather than environment customization.
- *Comprehensive benchmarking*: RL2Grid includes a variety of baseline algorithms, ranging from well-established RL methods to heuristic-guided approaches inspired by successful L2RPN solutions. This diversity ensures that RL2Grid is a robust tool for evaluating a wide range of RL strategies in power grid management. To our knowledge, RL2Grid is the first work to provide comprehensive learning curves for a suite of RL algorithms in these domains. This enables researchers to better understand the performance of different approaches and identify areas for improvement.

Specifically in regard to the points about the relationship with L2RPN and real-world power grids:
- *L2RPN*: Although the L2RPN challenge series (also built on top of Grid2Op) has offered valuable tasks to researchers, the solutions developed for these challenges often rely on highly customized actions, rewards, and heuristics that vary significantly between methods. This variability has made it difficult to perform meaningful comparisons and gain fundamental insights into the underlying factors driving performance differences. Notably, the L2RPN baselines cannot directly be implemented within the RL2Grid environments (given the differences in formalization), hence our approach of implementing baseline algorithms inspired by the innovations from the competitions but that are nonetheless comparable on common ground. On top of that, the L2RPN series shifted the focus of the competition at each edition, starting with testing the feasibility of developing realistic power network environments, to this year’s edition where the focus is on predicting the state of the grid (rather than controlling the grid). Every competition also comes with different time series, making effective comparisons and discovering important RL insights that are far from trivial.
- *Collaboration with power system operators*: RL2Grid was developed in collaboration with several power system operators. This partnership ensures that our benchmark is aligned with real-world power grid challenges, adding significant value to the RL research community. Our collaboration with system operators and experts in the field, also allowed us to identify and summarize the challenges related to using RL for power grid operations, which we believe will provide valuable insights to develop future research directions. This collaboration, along with an extensive preliminary experimental phase, has led to all our design choices for RL2Grid (e.g., reward design, goals, heuristics, types of actions, etc.).

**Withdrawal Confirmation:**

I have read and agree with the venue's withdrawal policy on behalf of myself and my co-authors.